# MRI and Adenomyosis: What Can Radiologists Evaluate?

**DOI:** 10.3390/ijerph19105840

**Published:** 2022-05-11

**Authors:** Veronica Celli, Miriam Dolciami, Roberta Ninkova, Giada Ercolani, Stefania Rizzo, Maria Grazia Porpora, Carlo Catalano, Lucia Manganaro

**Affiliations:** 1Department of Radiological, Oncological and Pathological Sciences, Policlinico Umberto I, Sapienza University of Rome, 00161 Rome, Italy; veronica.celli@uniroma1.it (V.C.); miriam.dolciami@uniroma1.it (M.D.); robertavalerieva.ninkova@uniroma1.it (R.N.); giada.ercolani@uniroma1.it (G.E.); carlo.catalano@uniroma1.it (C.C.); 2Clinica di Radiologia EOC, Istituto Imaging della Svizzera Italiana (IIMSI), 6900 Lugano, Switzerland; stefaniamariarita.rizzo@eoc.ch; 3Department of Maternal and Child Health and Urological Sciences, Policlinico Umberto I, Sapienza University of Rome, 00161 Rome, Italy; mariagrazia.porpora@uniroma1.it

**Keywords:** adenomyosis, magnetic resonance imaging, classification, endometriosis

## Abstract

Uterine adenomyosis is a common benign condition defined by the presence of heterotopic endometrial glands and stroma within the myometrium. Adenomyosis is often related to infertility and other adverse pregnancy outcomes. Modern imaging techniques allow the non-invasive diagnosis of adenomyosis and, in this framework, Magnetic Resonance Imaging (MRI) has assumed a central role due to its high diagnostic accuracy in the detection of adenomyosis. Currently, there is still a lack of international consensus on adenomyosis diagnostic criteria and classification, despite the fact that an agreed reporting system would promote treatment outcomes and research. This review aims to emphasize the important contribution of MRI to the diagnosis of adenomyosis and to highlight how, thanks to the great tissue differentiation provided by MRI, it is possible to identify the main direct (cystic component) and indirect (junctional zone features) signs of adenomyosis and to distinguish its various subtypes according to different MRI-based classifications. We also explored the main MRI criteria to identify the most common pitfalls and differential diagnoses of adenomyosis, whose features should be considered to avoid misdiagnosis.

## 1. Introduction

Adenomyosis is a common benign gynecological condition characterized by ectopic endometrial glands and stroma within the myometrium, with resulting smooth muscle cell hypertrophy and hyperplasia [1]. 

Adenomyosis is associated with a wide spectrum of clinical manifestations, ranging from asymptomatic to disabling conditions consisting of severe dysmenorrhea, dyspareunia, menorrhagia and menometrorrhagia and may be related to concomitant conditions such as anemia [2,3,4]. 

Moreover, recent studies have evaluated its association with infertility and other adverse pregnancy outcomes including elevated risk of miscarriage, preeclampsia and having a small child for gestational age [5,6]. Therefore, the identification of adenomyotic disease in young women is rising interest and requires valid non-invasive diagnostic methods. According to traditional estimates, above 20% of women who underwent hysterectomy for profuse uterine bleeding had a histological diagnosis of adenomyosis [1].

Nowadays, the exact etiology of adenomyosis is still unknown; however, the most widely accepted theories propose that endometrial glands directly invade the myometrium, resulting in angiogenesis of the spiral vessels and hyperplasia and hypertrophy of the adjacent smooth muscle tissue. This abnormal hypertrophic muscle tissue deepening into the myometrium prevents the regular uterine contractions that are responsible for coagulation in the spiral arteries; this mechanism appears to underlie menorrhagia or dysfunctional uterine bleeding. In contrast, other theories advocate misplaced pluripotent Müllerian embryonic remnants [7,8]. 

On the other hand, the scientific community agrees that estrogen exposure (short menstrual cycles, early menarche), parity, and prior uterine surgery (e.g., cesarean section, dilatation, curettage) are important risk factors [9,10].

As previously mentioned, the diagnosis of adenomyosis has long been of exclusive pathological relevance, and various definitions have been proposed over the years. Some of the most important authors who historically provided a histological definition of adenomyosis are Rokitansky in 1860, who defined adenomyosis as “fibrous tumors containing endometrial gland-like structures”; Cullen in 1920, who defined it as “endometriosis with predominant presence of fibromuscular tissue” and finally Sampons in 1921, who proposed a classification based on the origin of the adenomyosis, including subtype I, arising from the mucosal lining (invasion from within the uterus); subtype II, from the serosal surface (invasion from outside the uterus); and subtype III, arising from endometrial tissue misplaced in the uterine wall [11]. 

In the following decades, adenomyosis was not a subject of significant scientific progress in the diagnostic field, and its diagnosis remained for years exclusively dependent on surgical procedures with histological diagnosis. Currently, the histological diagnosis of adenomyosis is based on the identification of endometrial glands and stroma within the myometrium, at least 2.5 mm from the endometrial–myometrial junction, complemented by hyperplastic smooth muscle.

Recent advances in gynecologic imaging techniques achieved an increasing impact on the detection of adenomyosis, taking adenomyosis from a histologic diagnosis to a clinical entity and incorporating imaging criteria into the diagnostic workup [12,13]. The two main imaging techniques currently used are trans-vaginal ultrasound (TV-US) and magnetic resonance imaging (MRI), which represent a valid and accurate diagnostic tool to investigate young women with chronic pelvic pain or infertility [14,15,16].

As a widely available and relatively inexpensive technique, TV-US is the first option in the evaluation of adenomyosis. TV-US has high sensitivity and specificity when evaluating adenomyosis, corresponding to 0.72 (95% confidence interval [CI] 0.65–0.79) and 0.81 (95% CI 0.77–0.85), respectively [17].

In 2015, the international Morphological Uterus Sonographic Assessment (MUSA) group published a uniform and standardized reporting system of ultrasound findings for myometrial lesions, including adenomyosis [18]. The MUSA group defined some typical features of adenomyosis applicable in ultrasonographic diagnostic workup, including asymmetric thickening of the uterine wall, presence of intramyometrial cysts and/or hyperechogenic foci, fan-shaped shading of the myometrium, echogenic subendometrial lines and buds, translesional vascularization, and irregular or interrupted appearance of the junctional zone (JZ). Later in 2019, Van Der Bosch further explored this topic by drafting a US reporting classification which included disease location, distinction between focal and diffuse adenomyosis, identification of cystic/non-cystic elements, involvement of myometrial layer (Type 1: inner/sub-endometrial myometrium; Type II: middle myometrium; Type III: outer/sub-serosal myometrium), disease extension classified as mild, moderate, or severe, and measurement of lesion size [19].

The US examination can also benefit from other new diagnostic techniques. First, Color Doppler examination, which represents a useful tool to establish a differential diagnosis between adenomyosis and fibroids, often characterized by a higher signal on Color evaluation. Second, three-dimensional TV-US examination could provide a more accurate visualization of the JZ, especially in the coronal plane, although its ability to improve TV-US diagnostic performance has not been proven. Finally, another new US technique is represented by elastography that, through the different propagation of sound waves in tissues helps in the differential diagnosis between leiomyomas and adenomyosis. 

However, the main limitation is that ultrasound is an operator-dependent examination characterized by high variability between different exams; therefore, in cases of inconclusive US, MRI is considered the examination technique of choice.

MRI is considered a second-line investigation technique for adenomyotic disease due to its lower availability and higher cost compared to ultrasound. On the other hand, MRI is characterized by a higher a sensitivity (77%) and specificity (89%) and a lower operator dependence [20]. Moreover, MRI shows excellent soft-tissue differentiation, which allows the detection of other coexisting gynecologic conditions (e.g., fibroids or other manifestations of endometriosis), the identification of the JZ layer, the differentiation between various subtypes of adenomyosis and the evaluation of surrounding pelvic structures [21,22,23,24]. Several authors have proposed different classification systems for adenomyosis based on MRI features, although no international consensus has been reached [20,21,25,26].

Therefore, our aim was to highlight the important impact of MRI in the diagnosis of adenomyosis, showing how the excellent tissue differentiation of MRI lets identify the main direct (cystic component) and indirect (junctional zone features) signs of adenomyosis and to distinguish its subtypes according to different MRI-based classifications. We also explored the main MRI criteria to identify the major pitfalls and differential diagnoses of adenomyosis, whose features should be considered to avoid misdiagnosis.

## 2. Research Method

Relevant studies were identified using a computerized literature search in MEDLINE, Embase, and the Cochrane Library regarding articles published between 2000 and 2021. The MeSH Database of PubMed guided the search matching the following MeSH keywords: (adenomyosis) AND (MRI) AND (classification) 40; (adenomyosis) AND (MRI) AND (differential diagnosis) 69; (adenomyosis) AND (MRI) AND (pitfall) 14; (adenomyosis) AND (MRI) AND ((leiomyoma) OR (sarcoma)) 142. Among the articles obtained from our literature search, studies that provided a detailed description of the pathology and imaging methods were selected. Articles regarding adenomyosis treatment through imaging tools were removed; case reports and case series were excluded.

After an initial review of title, topic, or material and methods, articles that did not fulfil the scope of our review were excluded. Then, those documenting opinions, viewpoints, or anecdotes were removed.

Our initial literature research provided about 267 articles; subsequently 234 were articles eliminated based on the above criteria. Finally, 33 published articles were considered for this review.

## 3. Results

### 3.1. MRI Protocol

According to the European Society of Urogenital Radiology (ESUR) guidelines 2017, MRI should be considered the second-line examination technique for female pelvis disease after inconclusive TV-US investigation; moreover, it is recommended for endometriotic disease, for its accurate preoperatory staging [22].

Considering the central role that MRI plays in the diagnosis of endometriosis, ESUR recently developed a standardized MRI protocol for endometriotic disease, which can also be applied to the evaluation of adenomyosis.

According to the ESUR guidelines 2017, no significant differences were observed between final diagnoses obtained using a 1.5 or a 3T magnet, although on the basis of the better quality of the images, the use of a 3T magnet is preferred. Fasting 2–3 h before the examination and administration of an anti-peristaltic agent just before MRI scanning is recommended [22]. Bowel preparation is supported as “best practice”, particularly relevant for deep infiltrating endometriosis (DIE) detection, and consists of an enema 4–6 h before the acquisition, accompanied by a low-fiber diet for 2–3 days prior to the scan [22]. Moderate bladder filling is also recommended, as it avoids detrusor contractions generated by an overfilled bladder and the difficult visualization of ureters caused by an empty bladder. Therefore, in order to achieve moderate bladder filling, patients should be instructed not to urinate for at least 1 h prior to the MRI scan [23]. 

The supine position, entering “feet first” in the gantry, is also recommended, while the prone position may be considered for claustrophobic patients. The MRI imaging protocol (Table 1) for the study of endometriosis, according to the European Society of Urogenital Radiology, should include: High-Resolution T2 non-fat-saturated sequences on sagittal and axial planes (oblique plane if required; grade B recommendation); T1 non-fat-saturated sequences; T1 fat-saturated sequences. Contrast-enhanced fat-saturated T1 sequences are not mandatory for the diagnosis of endometriosis or adenomyosis, but are recommended in case of T2WI atypical features, to enhance mural tissue or adnexal masses, which may represent malignant lesions [22]. There was no recommendation for the use of DWI sequences. 

### 3.2. Junctional Zone: Physiological Changes

The junctional zone represents the inner myometrial layer, which exhibits different morpho-structural and functional characteristics compared to the outer myometrial layer. The excellent tissue differentiation provided by MRI allows distinguishing the JZ from the remaining parts of the uterus and identifying any morphologic changes that may lead to adenomyotic disease. This is particularly relevant for T2W sequences that allow the visualization of the trilaminar uterine structure composed of the inner and hyperintense endometrial layer, the intermediate and hypointense layer, representing the junctional zone, and the outer layer with intermediate signal intensity, representing the outer portion of the myometrium [20,21,25,26]. 

Nowadays, there is still a lack of consensus on both the thickness of the physiological JZ and the conventional threshold value for diagnosing internal adenomyosis by noninvasive imaging methods. Some studies have proposed 5–8 mm as a reference value for normal JZ thickness, whereas values between 8 and 12 mm indicate JZ hypertrophy [20,21,27]. Consistent with this, other studies have stated that a JZ thickening greater than 12 mm is highly predictive of adenomyosis, while a value less than 8 mm is consistent with normal JZ thickness [15,17].

When approaching the MRI examination, it is important to consider that the JZ undergoes thickness changes (cyclic and non-cyclic) that are mainly due to its function and its hormone-dependent origin. Although the JZ is anatomically a part of the myometrium, it has a Mullerian embryologic origin, similar to the endometrium, while the remaining portions of the myometrium have a mesenchymal origin.

Several factors are responsible for these thickness variations, which can lead to an overestimation of adenomyotic disease if not considered. First, the female reproductive cycle represents the most important factor in JZ changes, showing cyclic thickness variations concordant with that of the endometrium, due to their common hormone-dependent origin [28]. Particularly, during the menstrual phase the physiological pseudo-thickening of the JZ could falsely lead to a misdiagnosis of adenomyosis, which is why some authors suggest to avoid scanning during the menstrual phase [26].

Second, the JZ varies according to patient’s age, reaching its greatest thickness between 20 and 50 years, followed by a decrease in the postmenopausal period. In addition, in postmenopausal women, distinguishing between the JZ and the outer part of the myometrium is often difficult; in fact, fibrous involution of the outer myometrium causes a lower signal intensity of the T2W sequences of the latter, which appears similar to JZ’s signal intensity. Third, other hormonal conditions such as pregnancy and pre-menarcheal age may also make the JZ poorly identifiable. Moreover, oral contraceptives and gonadotrophin-releasing hormone analogs decrease the JZ thickness, while the non-steroidal anti-estrogen tamoxifen, used for breast cancer treatment, increases the incidence of adenomyosis in post-menopausal women [27].

The hormonal variations of the female reproductive cycle also influence the functional role of the JZ; in fertile women, uterine contractions physiologically are directed from the cervix to the fundus in the late follicular phase and from the fundus to the cervix in the late luteal phase [20,21]. Alterations in their generation and propagation could be the cause of uterine injury underlying JZ thickening in women with adenomyosis [29].

### 3.3. MR Classifications

Although for several years histologic diagnosis was considered the gold standard for the evaluation of adenomyosis, currently, MRI provides excellent tissue differentiation to distinguish the subtypes of adenomyosis. 

Over the past two decades, several authors have proposed different MRI classifications without reaching consensus on an accepted standard. In an early paper published in 2008 by Gordts et al., three main patterns for adenomyosis-related lesions were identified: (1) JZ hyperplasia, in the case of partial or diffuse JZ thickening ≥8 mm but <12 mm on T2-weighted images in women aged 35 years or less; (2) adenomyosis, in the case of JZ thickness ≥12 mm, presence of myometrial foci with high signal intensity on T2-weighted images and involvement of <1/3 or <2/3 or >2/3 of the external myometrium; (3) adenomyoma, in the case of an ill-defined mass included in the myometrial wall, showing a predominantly low signal intensity on all MRI sequences [27].

Kishi et al. in 2012 reported a new MR classification of adenomyosis, which was divided into four categories: subtype I, represented by the exclusive thickening of the uterine subendometrial layer without involving the outermost myometrium; subtype II, located in the uterine subserosal layer without involving the internal layers; subtype III, located in the middle myometrium and not involving internal or external structures; and subtype IV that includes adenomyosis not fulfilling the previous criteria [25]. 

In 2018, a new work by Gordts et al. suggested five parameters that should be defined for a complete MRI classification of adenomyosis: (1) affected area: internal or external myometrium; (2) location: anterior uterine wall, posterior uterine wall or fundus; (3) pattern: diffuse of focal; (4) type of lesion: muscular or cystic; (5) lesion volume or size: expressed either as <1/3, <2/3, >2/3 of the uterine wall or in cm [20].

Further improvement was achieved by the new classification based on MRI features proposed by Bazot et al. in 2018, that distinguished between internal adenomyosis, external adenomyosis and structure-related adenomyoma subtypes, and clarifying MRI diagnostic criteria for these different forms [26]. 

More recently, Kobaiashi et al. in 2020 proposed a new classification that considers all the main features of the previous ones. Specifically, they selected five main parameters: (1) the affected area, classified into internal myometrium (internal adenomyosis; the inner one-third of the uterine wall) or external myometrium (external adenomyosis); (2) the pattern, indicated as diffuse or focal adenomyosis; (3) the size, further categorized into three volumes (<1/3, <2/3, or >2/3 of the uterine wall); (4) the location, whether anterior, posterior, left-lateral, right-lateral, or fundal; (5) the presence of concomitant pathologies (none, peritoneal endometriosis, ovarian endometrioma, DIE, uterine fibroids or others) [21].

#### 3.3.1. Internal Adenomyosis 

Internal adenomyosis is characterized by ectopic endometrial glands and stroma displaced in the internal myometrium, with the resulting hypertrophy and hyperplasia of the adjacent smooth muscle cells. Typically, internal adenomyosis occurs in older women with severe menstrual bleeding and a prior history of uterine surgery.

The diagnosis of internal adenomyosis on MRI is based on the identification of direct and indirect signs. A direct diagnostic criterion is the identification of ectopic endometrial glandular and stromal components displaced in the inner myometrium, which, on MRI, appear as tiny cysts (2–9 mm) mostly characterized by high signal intensity in T2W images and sometimes by high signal intensity in T1W images due to hemorrhagic glandular content. 

The identification of indirect criteria is essentially based on JZ thickening, which represents a very controversial topic. Although these criteria are still elusive, the international literature agrees that MRI allows excellent tissue differentiation and the identification of uterine stratification. As already mentioned, many authors identify the T2W sequence as the most accurate in differentiating the JZ (hypointense middle layer) from the internal endometrial layer (hyperintense) and from the external myometrial layer (intermediate signal) [30]. Some authors also propose the T1 fat-saturated sequence as a useful tool to identify uterine trilaminar stratification, even obtaining better performance than that achieved with the T2W sequence [26].

For many years, a JZ thickness >12 mm measured on sagittal T2W sequence was considered diagnostic of adenomyosis [28,31]. However, many studies have reported conflicting values for the sensitivity and specificity of a JZ thickness >12 mm used as a diagnostic threshold value for adenomyosis (93% and 91% according to Reinhold et al. vs. a lower sensitivity of 63–70% and a specificity of 88–96% reported by two recent prospective studies) [2,15,16]. This may be explained by the variation in JZ thickness due to hormonal factors and the occurrence of sporadic uterine contractions and leiomyomas that make the JZ poorly visualizable. 

Therefore, recent literature has expressed the need to add other indirect criteria for adenomyosis, considering the thickness of JZ > 12 mm alone not sufficient for its diagnosis [15].

The first additional indirect parameter is the so-called “ratio_max_” representing the ratio between JZ_max_ and the entire myometrial thickness, both measured at the same point in the mid-sagittal plane; a ratio_max_ > 40% was found to have a sensitivity of 65% and a specificity of 93% for internal adenomyosis. A second parameter is the “JZ differential” (JZdiff), defined as the differential in maximal and minimal JZ thickness in both anterior and posterior uterine walls; the literature reports that a JZdiff > 5 mm has a sensitivity and a specificity of 70% and 85%, respectively [16]. Finally, the presence of a large smooth uterus shows low sensitivity (23%) for the presence of internal adenomyosis but is highly specific (98%) [14].

The literature suggests that none of these indirect criteria alone can be considered pathognomonic of adenomyotic disease; in fact, the simultaneous presence of these criteria is recommended for the diagnosis of this disease [26,32]. Moreover, a recent meta-analysis in 2021 proposed sensitivity and specificity values for the indirect and direct criteria similar to those previously mentioned [33].

In addition, the recent classification proposed by Bazot et al. (2018) distinguished three other subtypes of internal adenomyosis, potentially relevant to therapeutic management, namely, focal, superficial, or diffuse pattern. The focal type is characterized by small intramyometrial cysts with or without JZ swelling, distributed in single or multiple foci (Figure 1). 

The superficial type is represented by small diffuse cysts disseminated in the inner myometrium without JZ swelling (Figure 2). 

In addition, the superficial type can be defined as asymmetric if adenomyosis is predominant in only one of the uterine walls, or symmetric if it is equally distributed on both the anterior and the posterior walls. Finally, the diffuse type is characterized by small diffuse cysts in the inner myometrium combined with hypertrophic JZ; symmetrical and asymmetrical distribution are also evaluated (Figure 3).

#### 3.3.2. External Adenomyosis 

External adenomyosis is localized in the outer myometrial layers, involving the serosa and sparing the JZ [34]. It is often found in younger, nulligravid patients and frequently associated with deep endometriosis from which it seems to originate [25,35,36]. 

External adenomyosis has been classified into posterior and anterior subtypes by Bazot et al., based on its location.

It is most frequently localized in the posterior myometrial wall, where it is associated in 90% of the cases with DIE of the posterior compartment; only in 8% of the cases, external adenomyosis occurs in the anterior wall, usually combined with anterior DIE [37,38]. On MRI, both sites appear as a bulky irregular thickening of the subserosal myometrium, hypointense on T2W sequences and barely distinguishable from fibro-endometriotic plaques, when DIE is coexistent. In some cases, small internal cysts (2–9 mm) can be visualized as hyperintense elements on T2W sequences; if blood is present, the cysts appear hyperintense on T1W with and without fat saturation, a highly specific sign for the diagnosis of adenomyosis (Figure 4).

#### 3.3.3. Adenomyoma 

On MRI, adenomyoma presents as a heterogeneous lesion contained within the myometrial wall, not involving the JZ and the uterine serosa. On T2W sequences, it presents as a hypointense mass with ill-defined margins, showing internally small high-intensity cystic components or hemorrhagic cystic cavities >5 mm (hyperintense on T1W images) (Figure 5 and Figure 6) [39]. 

MRI is extremely accurate in discriminating the composition and location of an adenomyoma, features widely considered in the classification of Bazot et al. (2018), according to which three subtypes can be identified: (1) *Intramural adenomyoma*, represented by a poorly defined mass enclosed in the myometrial wall, which is in turn classified into two subgroups based on its content, i.e., *solid intramural adenomyoma*, containing only small cystic elements (hemorrhagic or not), and *cystic intramural adenomyoma* containing a hemorrhagic cystic cavity; (2) *Submucosal adenomyoma*, represented by a myometrial mass with poorly defined borders with small cystic elements which projects into the endometrial cavity; (3) *Subserous adenomyoma*, represented by a poorly defined mass with small cystic elements in the subserosal area.

### 3.4. Pitfalls

For a proper interpretation of MRI images, it is necessary to be aware of some pitfalls, consisting in the presence of elements that may mimic the adenomyotic disease. These pitfalls are mostly due to the hormone-dependent nature of the JZ, which influences its thickening. 

The female reproductive cycle represents the most important determinant of changes in JZ thickness, which peaks between days 8 and 16 of the menstrual cycle, mimicking adenomyosis [1,28]. For this reason, some authors suggest to perform scans in the late proliferative phase, avoiding the menstrual phase [15].

In addition, oral contraceptives, gonadotropin-releasing hormone analogs, pregnancy and postmenopausal status may reduce JZ thickness, which may be undetectable in about 30% of patients, reducing the MRI sensitivity for diagnosing adenomyosis.

As mentioned before, transient uterine contractions originating from the JZ are influenced by the menstrual cycle phase, proceeding from the cervix to the fundus in the late follicular phase and from the fundus to the cervix in the late luteal phase. On the T2W sequence, the uterine contractions are seen as small hypointense bands perpendicular to the JZ or as focal JZ thickening, which may mimic adenomyosis (Figure 7). After a few minutes, these transient focal findings disappear; therefore, a subsequent T2W image without them is sufficient to diagnose a JZ contraction [40]. In addition, the administration of hyoscine or anti-peristaltic drugs can be useful to reduce uterine contraction [41,42]. 

Lastly, adenomyosis may simulate an endometrial carcinoma through the so-called “pseudoenlargement of the endometrium” sign [43]. This is represented by hyperintense linear striations on T2W images, radiating from the endometrium toward the myometrium and giving the false impression of endometrial enlargement with myometrial invasion, which may result in the misdiagnosis of an endometrial carcinoma (Figure 8) [33].

### 3.5. Differential Diagnosis

#### 3.5.1. Leiomyoma 

Leiomyomas represent the most common and elusive differential diagnosis of adenomyoma.

Although both adenomyoma and leiomyoma are characterized by low signal intensity on T2W images, adenomyoma presents on MRI as a poorly defined lesion, with minimal mass effect and some tiny cystic components hyperintense on T2W or T1W sequences. In contrast, leiomyoma typically appears as a well-defined mass often associated with peripheral large vessels, usually not present around the adenomyoma. The differential diagnosis is even more misleading for leiomyomas in cystic or hemorrhagic degeneration that may contain central hyperintense areas on T1 and T2 images [1,28].

#### 3.5.2. Accessory Cavitated Uterine Mass (ACUM)

Accessory and cavitated uterine mass (ACUM) is a rare Müllerian uterine anomaly characterized by the presence of a cystic/hemorrhagic mass not communicating with the uterine cavity and enclosed in the myometrial wall [44]. ACUM is usually located near the origin of the round ligament; it appears as an accessory uterine-like mass, histologically composed of hemorrhagic chocolate cysts surrounded by an internal functional endometrium and an external layer of uterine musculature. It represents a difficult differential diagnosis of “intramural cystic adenomyoma” due to its clinical and radiological features [45].

On MRI, the diagnosis of ACUM requires the presence of a T1-hyperintense cavitated mass (due to blood content), completely isolated from the endometrial cavity. This round-shaped formation is outlined by three distinguishable concentric layers that coincide with uterine stratification and separate it from the endometrial cavity. This stratification from the inside out includes: a thin weakly hyperintense layer on T2W sequences showing moderate enhancement after contrast administration, corresponding to the functional endometrial layer; externally, a 3–8 mm hypointense layer on T1W and T2W, less enhanced than the myometrium after contrast medium, corresponding to the JZ; the outer part, a layer of normal myometrium with the same characteristics of the remaining myometrial portions [46]. ACUM differential diagnosis may include various gynecological conditions (rudimentary or cavitated uterine horns, intramural cystic adenomyoma and red degenerating leiomyomas); clinical–anamnestic data may also direct towards the correct diagnosis (Figure 9). In fact, ACUM is mainly detected in women under 30 years; therefore, in young women presenting severe dysmenorrhea and chronic cyclic pain, the diagnosis should be oriented towards ACUM or intramural cystic adenomyoma. In adult women, the differential diagnosis should primarily consider leiomyoma in red degeneration and cystic adenomyoma with hemorrhage content.

#### 3.5.3. MELF Endometrial Carcinoma

An additional and less common differential diagnosis is endometrial carcinoma with MELF growth pattern. In 2003, Murray et al. first described MELF tumors as characterized by microcystic, elongated, and fragmented glands with a fibromyxoid stromal reaction [47]. A MELF tumor is generally associated with myometrial and lymph vascular invasion and lymph node metastasis but a low histologic grade (G1 or G2). On MRI, it may present as T2-hypointense thickening of the inner part of the myometrium characterized by the presence of a tiny cystic component that mimics the direct and indirect signs of adenomyosis (Figure 10). MRI also shows the best diagnostic accuracy (83–92%) in the evaluation of myometrial invasion and allows the identification of pathological lymph nodes [48].

#### 3.5.4. Low-Grade Endometrial Stroma Sarcoma (LG-ESS)

Another malignancy should be considered among the possible differential diagnoses of adenomyosis: low-grade endometrial stroma sarcoma (LG-ESS). LG-ESS is an uncommon malignant mesenchymal tumor (<2% of all uterine malignancies), mainly detected in women under 50 years; it generally arises from the endometrium, showing an infiltrative growth patter into the myometrium, but in some cases, it may occur directly in the myometrium.

LG-ESS presents as an ill-demarcated infiltrating myometrial mass with uterine enlargement; on T2W images, it shows a heterogeneous intermediate signal with internal low-signal bands representing areas of infiltrated myometrium (Figure 11) [49]. Restricted diffusion and a high choline peak on MRI spectroscopy are typical features of LG-ESS, useful for its differential diagnosis. It is associated with perivascular infiltrative growth that may be better visualized on an ADC map [50].

## 4. Conclusions

In the last two decades, the advancement of noninvasive imaging methods has allowed the diagnosis of adenomyosis without the need of a surgical–histopathological procedure; this has provided physicians with a growing perception of both the true incidence of adenomyosis in the general population and its clinical manifestations. Although there is no international consensus on radiological classifications, the classification proposed by Bazot et al. in 2018 is the most widely accepted, attempting to provide a shared and uniform language to specialists.

However, in agreement with Kobayashi et al. (2020), other parameters should be included to improve this classification system [21]. It would be relevant, first to specify the location of each subtype of adenomyosis, describing whether it involves the anterior, posterior, right or left lateral wall, or the fundus. Second, to estimate the proportion of the uterine wall involved by adenomyosis (1/3, 2/3, or 3/3 of uterine wall). A further advance in the diagnostic work-up would be to highlight which symptomatic manifestations correspond to each subtype of adenomyosis on MRI, in order to better define its prognosis and improve its detection. However, no scientific evidence linking symptom severity to prognosis has been highlighted to date, probably because of the wide variety of interindividual manifestations (from asymptomatic to disabling) related to adenomyosis. The ultimate goal of a multidisciplinary collaboration would be the development of a shared classification system based on risk factors, clinical manifestations, and radiological features capable of directing toward the best possible treatment (surgical, medical, ablation with radiofrequency/microwave or focused ultrasound, HiFu).

## Figures and Tables

**Figure 1 ijerph-19-05840-f001:**
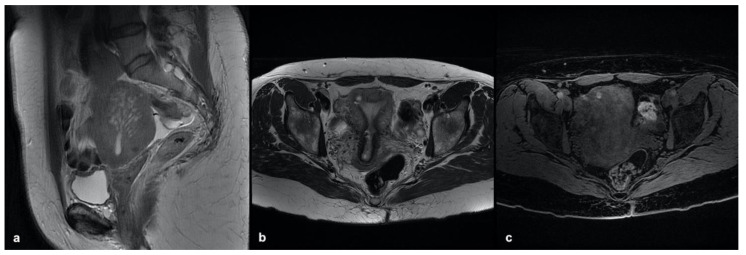
Focal internal adenomyosis. (**a**) Sagittal T2-weighted image showing a sub-endometrial tiny cyst with high-signal-intensity areas and JZ hypertrophy in both anterior and posterior uterine fundus. (**b**) Axial T2-weighted image showing an isolated sub-endometrial cyst characterized by both high signal intensity on T2- and high signal intensity on T1-weighted image (**c**) due to the hemorrhagic content.

**Figure 2 ijerph-19-05840-f002:**
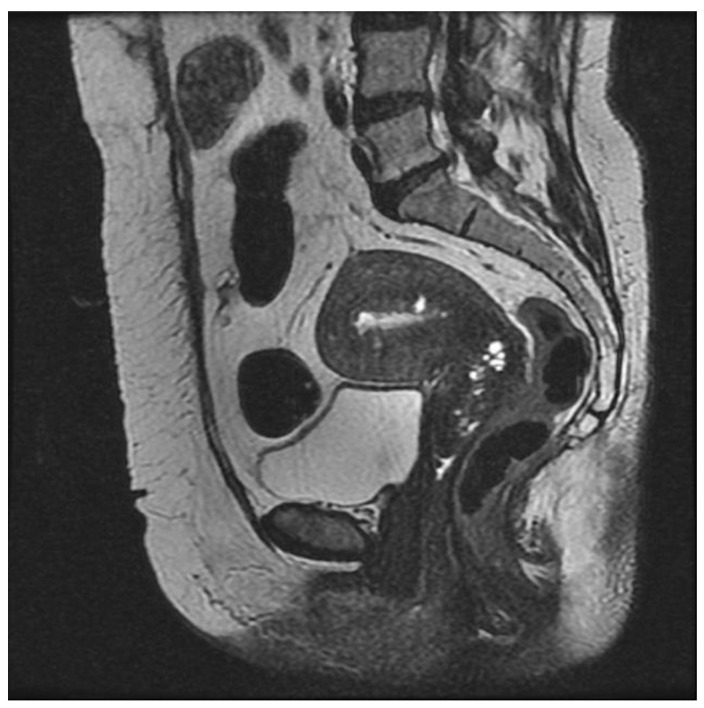
Superficial symmetrical internal adenomyosis: sagittal T2-weighted image representing a disseminated sub-endometrial tiny cyst without JZ hypertrophy in both anterior and posterior uterine wall.

**Figure 3 ijerph-19-05840-f003:**
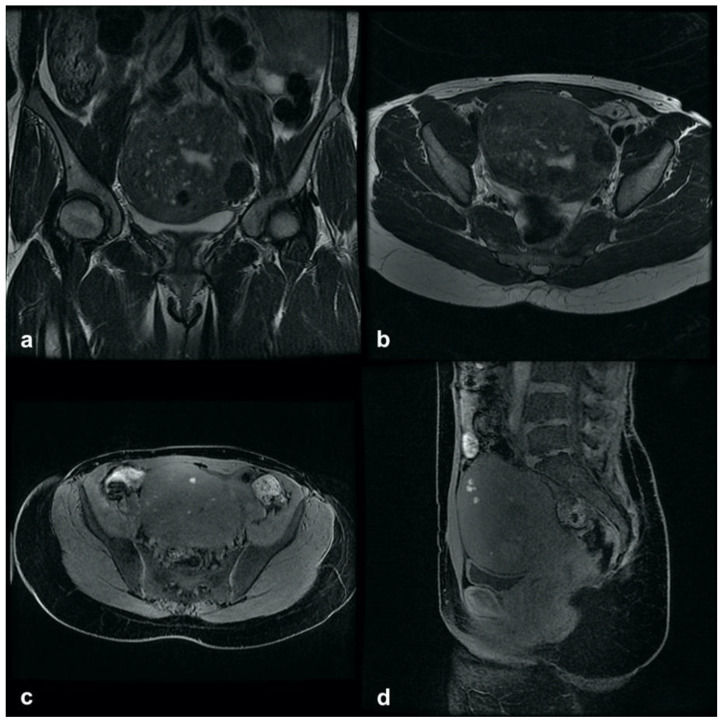
Diffuse asymmetrical internal adenomyosis: (**a**,**b**) Coronal and axial T2-weighed images showing an enlarged uterus with irregular thickening of the JZ, which appears as an ill-defined area, hypointense on T2W localized in the anterior myometrium. The lesion contains small high-intensity myometrial foci which represent ectopic endometrial tissue and small cysts. Two leiomyomas are also visible on the anterior and left lateral uterine wall as hypointense masses with well-defined borders. (**c**,**d**) Sagittal and axial T1-weighted fs image showing high-signal-intensity spots representing hemorrhagic content of the ectopic endometrial tissue.

**Figure 4 ijerph-19-05840-f004:**
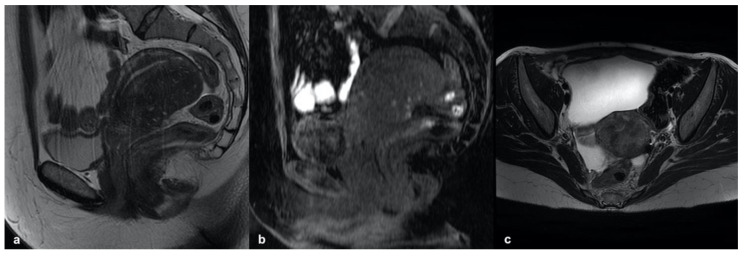
Posterior external adenomyosis: (**a**) sagittal T2-weighed image showing an hypointense ill-defined subserosa mass in the posterior myometrium and high-intensity cystic components within it. (**b**) Sagittal and axial T1-weighted image showing high-signal-intensity spots which correspond to some of the small high-signal-intensity areas seen on the T2-weighted image. This hyperintense foci on T1W represent hemorrhage within the ectopic endometrial tissue. (**c**) Axial T2-weighted image showing posterior external adenomyosis with deep endometriosis of the posterior compartment (involvement of the serous and muscular layers of the superior rectum showing hypointense thickening).

**Figure 5 ijerph-19-05840-f005:**
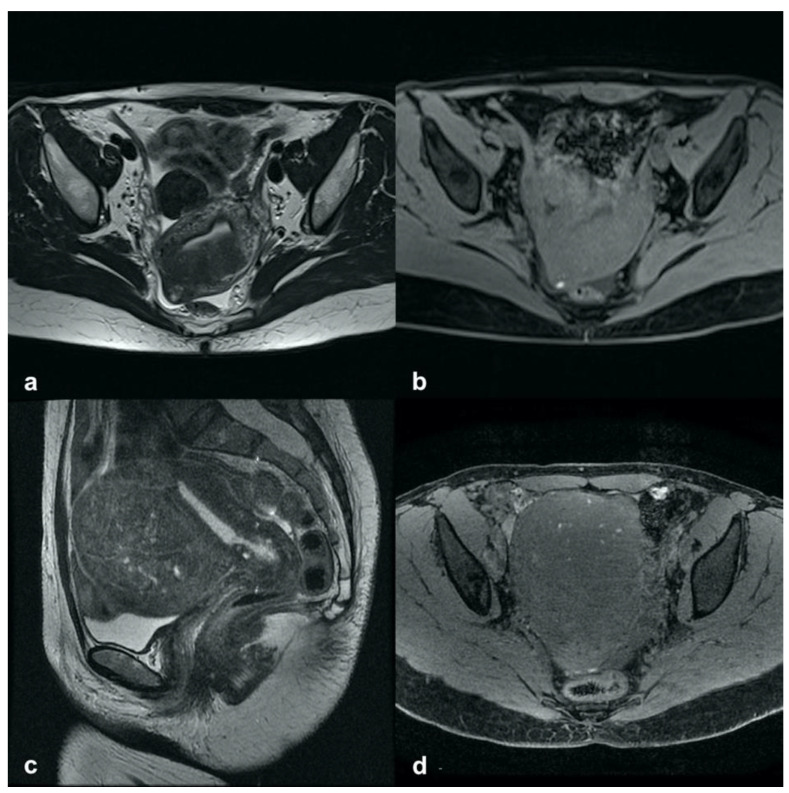
Adenomyoma: (**a**,**b**) subserosa adenomyoma with a small cystic component and hemorrhagic content. (**c**,**d**) Intramural solid adenomyoma: sagittal T2-weighted image showing an intramyometrial hypointense mass with ill-defined margins, minimal mass effect and tiny hyperintense cystic elements; axial T1-weighted image (**d**) showing high-signal cystic components within the adenomyoma, representing the ectopic glandular hemorrhagic content.

**Figure 6 ijerph-19-05840-f006:**
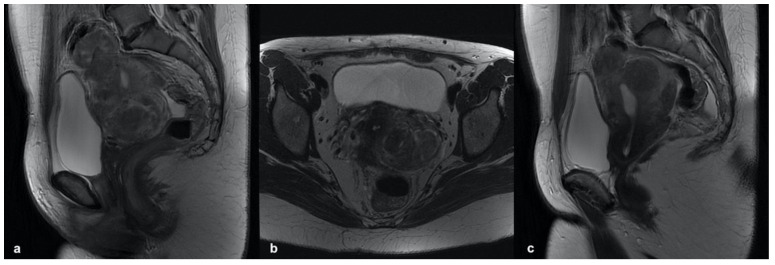
A 41-year-old woman affected by a different subtype of adenomyosis: (**a**) sagittal T2- weighted image showing anterior external adenomyosis with deep endometriosis of the sigma wall and a heterogeneous hypointense mass with well-defined borders, localized on the left wall of the uterine cervix ((**b**), axial T2) representing a subserosal leiomyoma. (**c**) Sagittal T2-weighted image showing an intramural solid adenomyoma of the posterior wall and posterior external adenomyosis.

**Figure 7 ijerph-19-05840-f007:**
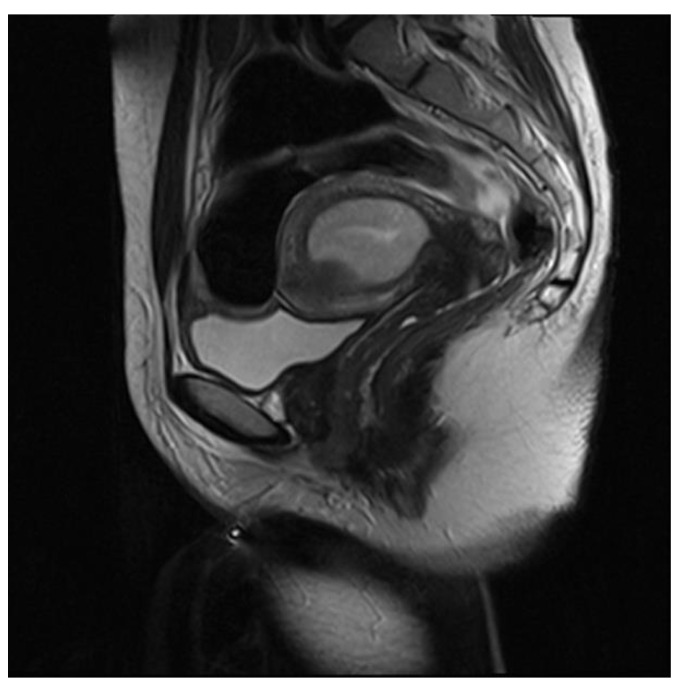
Physiologic transient myometrial contraction. Sagittal T2-weighted image showing focal low-signal-intensity bulging of the myometrium that mimics adenomyosis. This finding disappeared on subsequent T2-weighted images.

**Figure 8 ijerph-19-05840-f008:**
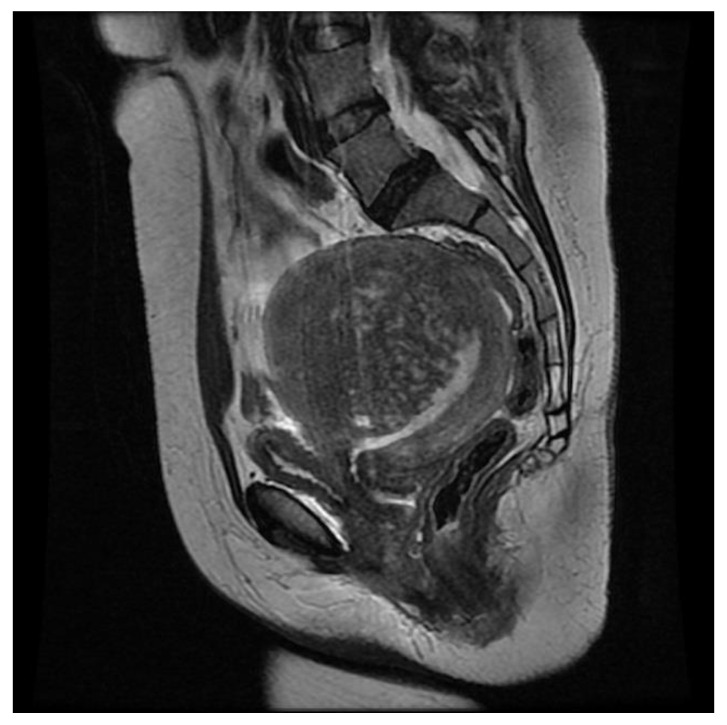
Pseudo-widening of the endometrium: sagittal T2-weighted image showing an asymmetrically thickened junctional zone (diffuse adenomyosis) with striated high-signal-intensity areas radiating from the endometrium toward the myometrium, a feature that simulates invasion by an endometrial carcinoma.

**Figure 9 ijerph-19-05840-f009:**
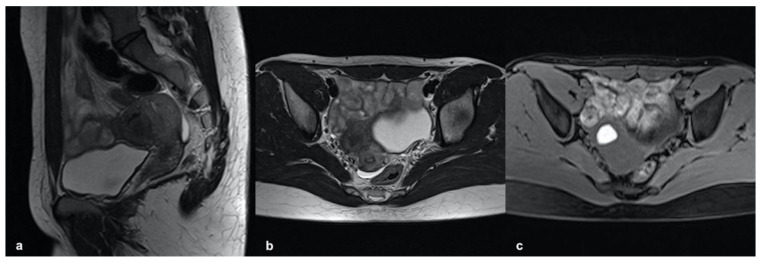
Intramural cystic adenomyoma: sagittal and axial T2-weighted images (**a**,**b**) showing a myometrial nodular mass with a central hyperintense cavity on T1-weighted images (**c**) due to the cavity’s hemorrhagic content and without connection to the endometrial cavity.

**Figure 10 ijerph-19-05840-f010:**
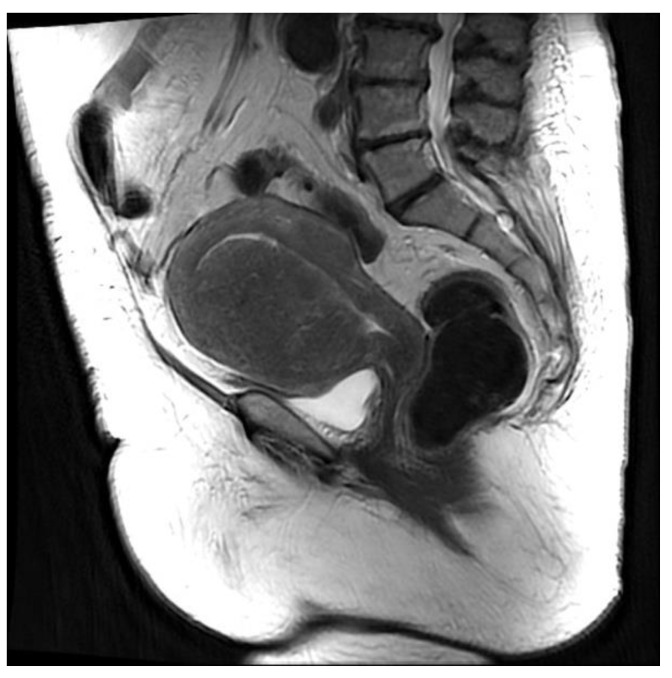
MELF endometrial carcinoma. Sagittal T2-weighted image showing the thickening of the inner part of the anterior myometrium and a low-signal-intensity adenomyosis-like mass with tiny cystic components.

**Figure 11 ijerph-19-05840-f011:**
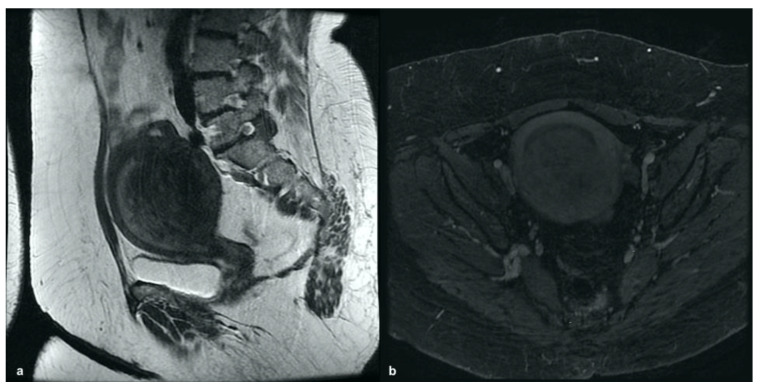
LG-ESS. (**a**) Sagittal T2-weighted image showing a low-signal-intensity adenomyosis-like mass in the posterior wall, which presents an heterogenous enhancement after the administration of Gadolinium, (**b**), contrast-enhanced fat-saturated T1-weighted image.

**Table 1 ijerph-19-05840-t001:** MRI Protocol. FOV: Field Of View; TSE: Turbo Spin Echo; TE: Echo Time; TR: Repetition Time; CE: Contrast Enhanced; FS: fat-suppressed; VIBE: Volumetric Interpolated Breath-hold Examination.

Sequence	TR (msec)	TE (msec)	FOV (mm)	Matrix	Thickness (mm)
** *T2 TSE* **	3000	68	260 × 260	320 × 256	4
** *T1 TSE with and without FS* **	500	Min	320 × 280	192 × 256	4
** *Non-CE and CE T1 VIBE* **	150	Min	320 × 280	256 × 256	4

## Data Availability

Not applicable.

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
