# Peer review of "MRI and Adenomyosis: What Can Radiologists Evaluate?"

_ijerph, 2022, doi:10.3390/ijerph19105840_

Round 1

Reviewer 1 Report

Although the revision has improved the manuscript I still miss the relevance of sonography (especially 3D), comparison with MRI and strengths and limitations of both methods.

Author Response

Thank you for your appreciated revisions and corrections.

We made some changes from the previous paper and we added a new paragraph from line 93 to line104 as it is marked up using the Track Changes function on Word.

Reviewer 2 Report

The manuscript has been much improved and is in a nice condition.

Author Response

Thank you for your comments, we appreciated your gentle response.

This manuscript is a resubmission of an earlier submission. The following is a list of the peer review reports and author responses from that submission.

Round 1

Reviewer 1 Report

This is a nicely written review concerning MRI-diagnosis of adenomyosis.

The literature search and interpretation seem to be adequate.

The authors should discuss more in detail pros and cons of US versus MRI and especially the consequences of a detected adenomyosis as to my knowledge there is little data about potential therapies which question the use of expensive diagnostic methods.

Reviewer 2 Report

This paper is a review of MRI classification and pitfalls in adenomyosis. Basically, the first half of this paper is similar to that of BAZOT 2018 and lacks originality.

Reviewer 3 Report

I read with great interest the review 'MRI and Adenomyosis: classification, typical features and differential diagnosis. The paper is well written and represents a complete  review of the adenomyosis. The pathogenetic aspect of the pathology, the classification and the diagnostic possibilities by means of magnetic resonance imaging are summarized. 

The topic is interesting as adenomyosis has a recognized role on fertility and bleeding disorders. Unfortunately it is often underdiagnosed. MRI plays an important role in diagnosis. Unfortunately, to date, we do not have diagnostic tools that can be easily applied in the follow-up of patients. 
